# Youth Entrepreneurship in Germany: Empirical Evidence on the How, the Why, the How Many, the Who and the When

**Rolf Sternberg** [1] **and David Breitenbach** [2,*]

1  Institute of Economic and Cultural Geography, Leibniz University Hannover, Schneiderberg 50, 30167 Hannover, Germany
2  PatentRenewal.com ApS, Nørre Voldgade 96, 1, 1358 Copenhagen, Denmark
*  Correspondence: david.breitenbach7@gmail.com

**Abstract:** Youth entrepreneurship is an increasingly prominent aspect of entrepreneurship support policies, but there is surprisingly little relevant research-based empirical evidence. This research gap is particularly noticeable when it comes to the personal and contextual factors that steer young people's decision to start a business. Using statistically representative survey data from the Global Entrepreneurship Monitor for Germany, we apply logit regressions to determine the influence of 10 independent variables on the likelihood of starting a business. We distinguish between 18–24-year-olds and 25–64-year-olds as well as between founders and non-founders. Self-efficacy in entrepreneurial skills, fear of failure and gender are the strongest influencing variables for the person-related factors and knowledge of other founders for the contextual factors. For younger people, the formal level of education and the perception of local entrepreneurial opportunities do not play a role in the decision to start a business, whereas they are very important for older people. Our results suggest that start-up promotion policies should explicitly address the empirically proven factors of youth entrepreneurship instead of a 'one size fits all' policy for new businesses, regardless of the age of the founders.

**Keywords:** youth entrepreneurship; entrepreneurship; Germany; spatial context; demographics; entrepreneurship policies; Global Entrepreneurship Monitor (GEM)

## 1. Youth Entrepreneurship: Relevance, Research Gaps and Opportunities for Academic Research

Entrepreneurial activities in the form of new venture creation are widely interpreted as drivers of economic development and growth (Wong et al. 2005). This statement holds true for each geographical dimension at the supranational level (e.g., the EU), the national level (e.g., Germany) and the regional and local level (i.e., the subnational level) in particular (Feldman 2001). New ventures are created by founders (either independently or collaboratively) whose personal attributes may have a statistical, but occasionally also causal, relationship with the start-up propensity of an individual and the probability of the success of this venture (e.g., growth, survival) (Gartner et al. 2004). One of these personal attributes is the age of the individual when they decide to start a business—or apply for a job (Cucculelli and Micucci 2019; Lévesque and Minniti 2006).

The popular notion is that new ventures are mainly started by young people; indeed, this stereotype has some evidence in the primary global start-up region referred to as 'Silicon Valley'. The founders of leading tech firms were indeed quite young when they started their business such as Apple Inc. (two of the three founders, Steve Jobs and Stephen Gary Wozniak, were 21 and 25 years old, respectively, when they started the business in 1976) and Windhorst Electronics GmbH (founded by the German serial entrepreneur Lars Windhorst in 1993 as a 16-year-old schoolboy). However, this is the exception rather than the rule in developed countries. While in many developing countries young people are indeed more entrepreneurial than the older generation—given the age structure in

such countries, young people often comprise the largest age group among entrepreneurs in absolute terms—the situation in high-income countries is completely different. Here, the middle-age groups (35–44 and 45–54) are the most entrepreneurial cohorts, both in absolute and relative terms (GERA 2023). The age distribution of the total population in such countries differs significantly from those in developing countries, where people over 50 currently comprise the majority. In absolute terms, of course, this has an impact on the number of young entrepreneurs compared to older entrepreneurs. When it comes to the economic, technological and societal effects of the age of the founders, there are comparative advantages of either age group in terms of creativity, flexibility, (low) transaction cost and (sufficient) time for return on investment for young entrepreneurs as well as advantages in terms of work experience, start-up experience, capital and networks for older entrepreneurs (for Germany see, e.g., Sternberg 2019). In most high-income countries, the middle-age group shows the highest propensity to start a business, making an inverted U-curve a frequent diagram form of the probability to start a business during their lifetime (Guerrero et al. 2021).

Nevertheless, in addition to the more general evidence presented above, there are many open research questions regarding youth entrepreneurship, both from a theoretical and an empirical perspective. While there is some empirical evidence of youth entrepreneurship, it is mainly restricted to emerging/developing countries (see, e.g., Burchell and Coutts 2019; van der Westhuizen and Goyayi 2020; or Baluku et al. 2019), where starting a business often has a different economic and social relevance than in high-income countries.

When dealing with young entrepreneurs, most studies in the latter countries address only university students or graduates, who are not emblematic of all young entrepreneurs, no matter which country type is considered (e.g., Turker and Selcuk 2009; or Georgescu and Herman 2020). In particular, little is empirically known about the specific mechanisms, attitudes, competencies and motivations of young entrepreneurs compared to older entrepreneurs (Geldhof et al. 2014; see also the critique described in Maleki et al. 2021). The same is true in the case of theoretical explanations. Several of the established theoretical attempts to explain an individual's decision to start a venture (or to be employed) do *not* work for very young founders—or have at least not yet explicitly and systematically been tested for young founders (e.g., trait approaches such as the personality attributes; demographic attributes such as gender, migration background, precise age or start-up experiences; start-up motivations; skills and the like). Accordingly, we do not know whether (and how) these aspects have changed over time, e.g., whether they have changed during the COVID-19 pandemic in a way that, besides entrepreneurship, makes/made young people suffer in a particular way and at a certain level of intensity compared to older people (Boris et al. 2021; Gavriluță et al. 2022). This relative lack of theoretical and, moreover, empirical research on youth entrepreneurship is surprising given the strong correlation (also among policymakers and their effort to support young ventures and/or young entrepreneurs) between young ventures and their seemingly young founders.

In a nutshell, a combined view of theoretical and empirical academic work on youth entrepreneurship shows a relatively large number of publications on entrepreneurial intentions (e.g., Kaya et al. 2019; Shirokova et al. 2022) and fear of failure as an obstacle to starting a business, with young people being less risk averse, (Tubadji et al. 2021). Moreover, youth self-employment is a prominent aspect of research on youth entrepreneurship, often based on data from the UN or the International Labour Organization. However, demographic factors such as gender or education as well as institutional or contextual factors of the respective country or subnational region are under-researched topics in studies on youth entrepreneurship.

For some of the described empirical research gaps, the simple reason for youth entrepreneurship being an under-researched topic within the academic domain of entrepreneurship is the lack of primary internationally comparable and panel-like data. When it comes to survey data, the Adult Population Survey (APS) of the Global Entrepreneurship Monitor (GEM) has, by far, the largest and oldest international comparable database, providing the exact age (in years) at the time of the interview (for details on the APS questionnaires, see Reynolds et al. 2005; Bosma 2012; Bosma et al. 2012; and www.gemconsortium.org, accessed on 22 July 2019).

This paper's focus is on youth entrepreneurship in Germany. This country provides a particularly compelling case for studying these questions. Even compared to other high-income countries, Germany has a low overall entrepreneurship rate (GERA 2023; Sternberg et al. 2022) but a large number of well-educated young people and a strong overall economy (World Bank 2021). Moreover, the country suffers from a lack of qualified younger people (compared to the available jobs for such people), which is closely related to the overall age structure of the population (many older and fewer young people) and, consequently, of the employees in total. While the total population slightly increased in recent years due to immigration, this has not yet helped to solve the severe labour market problems. Hundreds of thousands of jobs cannot be filled because of a shortage of skilled workers in both well-paid jobs (industries such as software or mechanical engineering) and low-paid jobs (in industries such as health services and the care sector (Brenke 2022). One result is the power of the few: the relatively few very well-educated people in shortage occupations have a very good bargaining position and can negotiate extremely favourable working conditions. Given a traditionally not-very-entrepreneurial-prone society and the high opportunity cost of starting a business by highly qualified labour in Germany, the low overall start-up rate in Germany is a logical consequence.

However, the situation could potentially be different for young people. Young people's aspirations for their lives, careers, and finances change over generations and usually differ from those of their parents or grandparents. For example, younger people, at least in Germany, attach more importance (than older people) to the meaning of work, a lower daily and lifetime working time (which is also supposed to be very flexible) as well as the compatibility of work and family, while career and a high and regularly rising wage have become relatively less important (Deutsche Shell Holding GmbH 2019). This may also affect the decision in favour of starting a business or working as an employee. Indeed, a recent analysis published as part of the annual Global Entrepreneurship Monitor (GEM) Country Report Germany shows that younger people who are starting a business exhibit somewhat different entrepreneurial motivations than older entrepreneurs (see Sternberg et al. (2022) for the most recent GEM Country Report Germany), particularly regarding the fact that the new business 'should make a difference'.

Based on individual data from the Adult Population Survey (APS) of the GEM Country Report Germany for a sufficiently long period, we address four research questions. First, we measure in a descriptive part of our paper how frequent youth entrepreneurship is and has been during the last two decades in Germany (compared over time and across countries). Second, we analyse how the young founders themselves differ in terms of personal attributes from older founders. Third, we investigate how young founders differ from non-entrepreneurial young Germans in terms of entrepreneurial perceptions and attitudes (mirroring the context in which they live) as well as personal attributes. Finally, we provide some empirical evidence of the factors in young people's decision to start a business (instead of working for an employer) compared to the rest of the population. These factors are divided into personal attributes and contextual aspects.

In this paper, we provide the first empirical analysis and assessment of the prevalence of youth entrepreneurship in Germany and the relevant individual factors. Our study contributes to two highly relevant areas of research. First, this paper is an investigation into the amount of youth entrepreneurship in Germany that shares many entrepreneurship attributes with other high-income countries such as low start-up rate compared with low-

income countries, high impact of knowledge-intensive products for the country's global competitiveness, emerging attention of policymakers to entrepreneurship in general and of young people as the founders of such ventures in particular. At a time when poorer countries confront unprecedented increases in population and developed countries have ageing populations, we provide important insights into the relationship between demographic structure, aggregate entrepreneurship and growth (Lévesque and Minniti 2011). Second, our analysis addresses the important question (both from an academic and a government policy perspective) of which person-related and contextual factors have an impact on a young individual deciding to start their own venture instead of searching for a job as an employee. While there is a vast amount of literature on start-up decisions in general, the empirical evidence of youth entrepreneurship is quite scarce—particularly regarding the situation in Germany.

The remainder of the paper is organized as follows. In the next section, we describe the literature on youth entrepreneurship in general and in Germany, as well as the personal attributes and contextual aspects, and our conceptual model and related hypotheses are set out. Section 3 is dedicated to the presentation of our data, the variables and the methods used. The results of our empirical analysis are presented in Section 4. Section 5 discusses our findings, and Section 6 will focus on conclusions and limitations.

## 2. Literature Review: Theoretical Explanations for and Empirical Evidence of the Volume and the Micro-Level Factors of Youth Entrepreneurship

### 2.1. Youth Entrepreneurship—Definition, Evidence and Relevance

In the lengthy history of entrepreneurship research, most scholars have not focused their research efforts on the area of youth entrepreneurship. Geldhof et al. (2014) attempted to demonstrate how young this particular research area is by pointing out that there was not only a general lack of studies on youth entrepreneurship in 2014, but most reviews of entrepreneurship literature even failed to mention the significance of the topic. Perhaps stemming from the late maturation of this research area, there is also a lack of a common definition or understanding of the term 'youth entrepreneurship' (Holdsworth and Mendonça 2020). In simpler terms, while people might generally use the term 'youth' to describe the period of an individual's life when they are young, it is a highly subjective term that could vary across countries, cultures or even people's perceptions (UN 2023). To demonstrate this, this study found evidence in the existing literature that some studies might use the term 'youth' to refer to people between the ages of 15 and 20 (Hekman 2007), while others view youth in a much wider context such as between the ages of 15 and 35 (Schøtt et al. 2015; Maleki et al. 2021). However, most researchers define youth as the period of one's life when they are between the ages of 15 and 24 (Hulsink and Koek 2014; Burchell and Coutts 2019), sharing the UN's statistical approach to the topic (UN 2023). In our empirical paper, we define youth entrepreneurship as a phenomenon in which individuals embark on their entrepreneurial journey by initiating entrepreneurial activities between the ages of 18 and 24. This definition is mainly driven by the availability of data as described in Section 3.

In recent years, a positive tendency can be seen in the research activities in this area, partly because young people are increasingly starting businesses partly due to unemployment; thus, the relevance of this topic increases as well (Green 2013; Schøtt et al. 2015; Burchell and Coutts 2019; Maleki et al. 2021). The most recent OECD Employment Outlook (2022) shows that the average youth unemployment rate in 2022 was 15% and 13% in the EU and US, respectively (OECD 2022). There is also increasing evidence from developing countries on youth unemployment (see e.g., Baluku et al. 2019; Blattman et al. 2022; Geza et al. 2022); thus, the problem seems to be posing a significant global challenge. Upon analysing the same problem, Schøtt et al. (2015) found that the economic and social costs of youth unemployment are considerable, especially if people cannot find jobs for long periods. First, one potential consequence of their long-term unemployment might be that their skills will be eroded and their potential to enter the job market continuously decreases.

Second, the high levels of youth unemployment can lead to underutilized human capital and talent (Schøtt et al. 2015).

As a potential solution to this global challenge, studies have found that entrepreneurship might just provide youth with the appropriate avenue to enter the job market (Hofer and Delaney 2010; Schøtt et al. 2015; Boris et al. 2021; Gavriluță et al. 2022). Others have shown that youth entrepreneurship does not only provide the youth with an appropriate avenue for long-term value creation in the economy, but youth entrepreneurship can have a positive impact on the economy by creating jobs and driving innovation, thereby contributing to the overall competitiveness of local economies (Murithi 2013). Moreover, youth entrepreneurs will be in a position where they will inevitably need to develop their skills and knowledge to keep their business afloat, contributing to their personal growth as well (Schøtt et al. 2015; Maleki et al. 2021).

*2.2. Factors in Youth Entrepreneurship*

Given the described potential economic and political relevance of youth entrepreneurship, the factors that steer it become relevant. This paper will group the factors into two main parts: (1) person-related (or internal) attributes; and (2) contextual (or external) factors. The vast majority of literature on youth entrepreneurship focuses on internal factors such as entrepreneurial intentions (see Turker and Selcuk 2009; Kaya et al. 2019; Mawson and Kasem 2019; Swaramarinda et al. 2022) or demographic factors influencing an individual's entrepreneurial journey in their youth (see Hofer and Delaney 2010; Baluku et al. 2019; Boris et al. 2021; Djordjevic et al. 2021; Marchesani et al. 2022; Santos-Jaén et al. 2022). In this section, a selective discussion will be presented on the factors of entrepreneurship along with their associated variables and their relation to youth entrepreneurship.

Person-related factors influence one's intentions for business venturing studies and the eventual decision to start a business or not. Well analysed, these factors include human capital (including entrepreneurial skills), gender, household income, education and fear of failure as a reason not to start a business (Carland et al. 1988).

Empirical research shows that human capital has a significant influence on one's entrepreneurial intentions and decision to start a business venture. Most studies in this area unanimously agree that education can boost youth attitudes and interests in business venturing (Turker and Selcuk 2009; Geldhof et al. 2014; Maleki et al. 2021), while it also provides opportunities for gaining deeper knowledge of risk assessment, innovativeness and 'know-how' (Hsu et al. 2019). Regarding age differences, Schøtt et al. (2015) found that young entrepreneurs have more extensive education and training than older entrepreneurs. Based on GEM data, research shows that three-quarters of youth entrepreneurs have at least completed secondary education compared to two-thirds of older entrepreneurs, providing them with a higher chance of acquiring entrepreneurial skills.

GEM Global reports as well as GEM-based country reports of over two decades show that men in almost every country are involved in entrepreneurial activities more often than women (Martinović and Lakoš 2012; Schøtt et al. 2015; Pilkova et al. 2019). This is also true for Germany, which is the focus of the empirical core of our paper. Interestingly, gender is also considered an important factor in youth entrepreneurship, as most research papers acknowledge that there is a gender gap in the entrepreneurial activity of the youth (Schøtt et al. 2015; Vamvaka et al. 2020). Some studies suggest that an entrepreneurial career poses higher personal and social risks for young female entrepreneurs than for their male counterparts (Vamvaka et al. 2020). One potential reason for this is that female entrepreneurs might face negative implications in terms of social benefits related to family and children as they tend to have higher levels of domestic responsibility (Schøtt et al. 2015). Schøtt et al. (2015) have analysed other potential reasons for this phenomenon such as a general lack of female role models in the business sector, lower levels of education of young women in many developing countries, less business-oriented networks of women and lack of public support for female entrepreneurs (and young ones in particular) due

to less confidence in the success of female-led businesses. When comparing different age groups by gender, academic research fails to provide a country-specific analysis.

The literature on young entrepreneurs often deals with the perception of their own entrepreneurial skills as a factor in entrepreneurial intentions. There is strong empirical evidence that the youth more frequently have the intention to start a venture than adults or senior entrepreneurs, even though they have limited financial resources, life and work experience, and they face greater barriers of entry (Schøtt et al. 2015). It is important to distinguish between the objective availability of such skills and the individual perception of whether or not an individual possesses such skills. While the actual skills will influence the success of the start-up in the mid and long run, the perception of such skills strongly impacts the probability of start-up intentions becoming start-up activity, i.e., whether the business will be created or not. Several studies highlight that senior or adult entrepreneurs tend to have higher entrepreneurial self-confidence as well as a wider array of skills and experiences; thus, they might be more perceptive to entrepreneurial opportunities than the youth (Rehak et al. 2017; Pilkova et al. 2019).

Regarding household income, research shows that youth unemployment is one reason why young people are starting businesses (Green 2013; Schøtt et al. 2015; Burchell and Coutts 2019; Maleki et al. 2021). Unemployment directly correlates with low household income, and the economic and social costs of this phenomenon are considerable. In a comparison between senior and youth entrepreneurs in European countries, Pilkova et al. (2019) showed that the youth in developed countries tend to be discouraged from starting a business, as they rely on good employment opportunities; thus, good opportunities to collect a decent household income in turn decreases their self-confidence by providing them with a safe job opportunity. The same research shows that necessity-driven youth entrepreneurship (as opposed to opportunity-driven entrepreneurship) dominates, as the youth views entrepreneurship as a better career choice than being an employee (Pilkova et al. 2019).

Fear of failure can be defined as a general attitude of fear that encourages one to avoid risk and uncertainty or to take risks and dive into the unknown (Shapero and Sokol 1982; Wennberg et al. 2013; Adam et al. 2018). Interestingly, this factor of entrepreneurship can be considered both internal and external, as it is highly dependent on individual attitudes and behaviour (Tubadji et al. 2021), which have been proven to pass down through generations. Research has also found evidence that an individual's attitude towards risk is influenced by role models and the local cultural context (Wennberg et al. 2013). While there is a general risk-taking or risk aversion that stems from culture, a fear of failure can also be influenced by personality traits, societal norms and regulations. Certain demographics might have a different propensity towards risk-taking (Mather et al. 2012). Based on four experiments, younger adults have shown a weaker preference towards sure gains and a weaker risk aversion for sure losses than older adults, who took outcomes into consideration more heavily when a choice was offered between a certain option and a riskier option. Similar findings have been presented by Schøtt et al. (2015), who found that younger youth (18–24 years) have a slightly higher willingness to take on risk than older youth (25–34 years) and adults (35–64). One explanation is that young people in general have not gained lived experience in self-efficacy, opportunity alertness and risk aversion; instead, they have gained their competencies through their network, social bonds or education (Hoffmann et al. 2005).

Other person-related factors are not that well studied. For example, previous entrepreneurship experiences and/or supporting other new firms as a so-called 'business angel' may have an important impact on an individual's propensity to start another new business and/or to start their own business for the first time.

For the other main group of factors of youth entrepreneurship—the external or contextual ones—the literature mentions external incentives (Tubadji et al. 2021) as well as societal and cultural aspects (Geldhof et al. 2014; Maleki et al. 2021; Tubadji et al. 2021). Many empirical studies configure the context in a geographical sense, for example, by distinguishing between local, regional, national or supranational context (see Sternberg (2022) for a recent overview). The regional context is very often considered to be the most appropriate spatial scale to analyse the contextual factors of both entrepreneurial activities (the decision to start—or not start—a business) and entrepreneurial success (i.e., the survival rate or growth of new ventures). In a nutshell, 'Entrepreneurship is primarily a regional event' (Feldman 2001). Moreover, the temporal context might matter when explaining entrepreneurial activity. For example, Fritsch and Wyrwich (2019) have developed a complex socialist heritage hypothesis to explain the entrepreneurial differences between the Eastern (and former socialist) part of Germany and the Western part. All these contextual factors tend to influence someone's entrepreneurial actions. Contextual differences, especially in quantitative–empirical work and in a spatial interpretation of the context, are often described with the—somewhat imprecise—term 'entrepreneurial culture' (Hayton and Cacciotti 2013). Often, this term is merely a placeholder for non-personal attributes of a territory whose influence on individual entrepreneurship decisions is to be investigated. There is empirical proof that societal and cultural factors could all potentially influence one's intentions to start one's own business as well as influence entrepreneurship's desirability and feasibility (Maleki et al. 2021). Nowiński et al. (2020) found that the positive perception of public support indirectly influences one's entrepreneurial intentions because if someone's demonstrated attitude towards entrepreneurship is supported by the actions, beliefs or attitudes of society, they are likely to continue their actions. Public perception might also play a significant role in influencing the youth's self-efficacy (perception of feasibility) and risk attitudes (Nowiński et al. 2020). Other evidence also demonstrates that even if someone is motivated to start their business by financial gains or other means of achievements—whether it is desire for social success, career success or individual fulfilment—a national culture that encourages and supports entrepreneurial activities is needed (Lee and Peterson 2000). A culture that rewards and motivates venture creation can positively influence the youth's entrepreneurial intentions, especially if there is a cultural aspiration for excellence and competitiveness (Lee et al. 2005). Other studies also support the fact that successful entrepreneurs are created in entrepreneurship-oriented cultures and societies (Watson et al. 1998; Lee and Peterson 2000; Morrison 2000; Adam et al. 2018).

For these contextual factors, which are only briefly described here, there are several methodological options in empirical studies, which are exemplified by the spatial context. First, the data can refer to characteristics of the region in question and represent mean values for all inhabitants of this region, e.g., the GDP/per capita of Region 1 as an economic indicator of the region—undoubtedly an important variable that influences an individual's decision to start a business (Bosma 2012; Bergmann et al. 2016). The data for this variable then come from relevant official data sources such as the INKAR database in Germany. A second option is survey-based individual data from a sample of, e.g., the population (e.g., on the migration background or the gender of the respondents) or start-up experts (who, e.g., assess the entrepreneurial framework conditions of the territory) of the spatial unit in question, which are used in relevant regression analyses after aggregation and calculation of a mean value for the territory in question. For example, the GEM project, the largest and oldest international research consortium for the analysis of entrepreneurship activities worldwide, works in this way. A third option is used much less frequently, especially due to a lack of data, although it offers great potential, as it uses the perception of entrepreneurs or potential entrepreneurs (i.e., the population) to indirectly obtain an assessment of the spatial context—precisely from those individuals whose entrepreneurial activities and decisions are to be explained. This is the methodological approach of this paper (more on this in Section 3).

Contextual factors in the entrepreneurial activities of young people also include their perceptions (instead of real attributes such as the GDP per capita in the surrounding territory) of the geographical context in particular. Of course, individual perceptions of these factors may differ from their actual characteristics. However, it is the perception that is decisive for the actions of the individuals (in this case the young people), not the real environmental conditions. Therefore, there are also indications in the literature of how the perception of, for example, the reputation of entrepreneurs in the society in question, influences entrepreneurial propensity (Tubadji et al. 2021). The same applies to the frequency of media coverage of new businesses and young entrepreneurs (Tubadji et al. 2021) as well as the image of starting a new business as a career option. The perception of entrepreneurial opportunities as good (at present and in the region where the individual lives) also influences the individual's propensity to start a business. If an individual perceives these opportunities as poor, they are unlikely to start a new business anyway. This perception, which of course significantly determines the individual's actual actions, is, therefore, an important context variable (Schøtt et al. 2015). Previous analyses on this topic, however, did not distinguish between younger and older people. In addition, the size of the individual's network is a potentially significant factor in both the start-up decision and start-up success (Schøtt et al. 2015). In particular, networks have a compensatory function, as network partners can, at least partially, compensate for the lack of competence, experience and reputation of younger founders or the lack of customers for the new business (Schøtt et al. 2015). In addition, the network can include other former or current founders who can function as role models (Köllinger et al. 2005; Fritsch and Wyrwich 2019) and increase the propensity to found a business. The empirical results to date regarding such network and role model effects on the decision to start a business do not control for the age of the potential or actual founders.

Although this section has shown that there is clear evidence of the youth having high intentions to start their own businesses, studies have stressed the fact that most results have come from cross-cultural and transnational research. It is recommended that country-level studies be conducted to explore the exact differences on societal, cultural and demographic factors influencing the youth's entrepreneurial intentions (Sieger et al. 2019; Maleki et al. 2021). Our paper addresses this research gap by examining the status of youth entrepreneurship in Germany in an attempt to understand the current status of youth entrepreneurship, the factors in youth entrepreneurship and the cultural and societal factors that influence young people in Germany.

### 2.3. Youth Entrepreneurship in Germany—Conceptual Model and Hypotheses

In this subsection, we use the described current state of youth entrepreneurship in general—in Germany in particular—to illustrate our hypotheses and construct a conceptual model.

### 2.3.1. Internal Factors

This paper considers the following six separate person-related factors for its conceptual model:

- Gender;
- Household income;
- Education;
- Business angel activity;
- Perception of own entrepreneurial skills;
- Fear of failure (as a reason NOT to start a business).

The entrepreneurial intention in Germany generally seems favourable; Hekman (2007) has shown that German youth have a positive attitude towards entrepreneurs, as 99% of their survey participants considered entrepreneurs either 'favourable' or 'somewhat favourable'. Most participants either had entrepreneurs in their network or their parents or teachers might have transferred their positive attitudes towards entrepreneurs.

Connecting this with Hoffmann et al.'s (2005) study, we assume that German youth gain their entrepreneurial competencies from their network through social bonds or education. Education and social embeddedness also positively affect one's perception of one's entrepreneurial skills and influences one's propensity to start a business. As the annual GEM Country Report Germany have shown for several years—albeit usually without differentiation by age category—around 40–50% of adults in Germany believe that they have the skills required to start a business. This applies to a lesser extent to younger people (Sternberg et al. 2022). International comparisons of this perception of one's own entrepreneurial skills among 18–24-year-olds show that Germany is approximately in the middle of the field among high-income countries with this value (Schøtt et al. 2015). However, the sources mentioned do not examine the impact of this self-efficacy on the actual decision to start a business. When taking the human capital aspect into greater consideration, Hekman (2007) found that young Germans believe that they have inadequate knowledge of economics and entrepreneurship. Specifically, in his research, he saw only 8% of his participants consider their economic knowledge 'good', while more than 90% of the participants believed that they have 'some' or 'hardly any' economic and entrepreneurial knowledge (Hekman 2007). Studies have also demonstrated that education can significantly boost young people's attitudes and interests in business venturing (Turker and Selcuk 2009; Geldhof et al. 2014; Maleki et al. 2021), as well as provide a deeper knowledge of risk assessment, innovativeness and 'know-how' (Hsu et al. 2019). Thus, Germany might be in a similar position than other high-income countries examined in the cited studies, so we assume that the youth's perceived knowledge of their entrepreneurial skills is a more significant factor in the start-up decision than it is for older people.

**Hypothesis 1.** *The perception of one's own entrepreneurial skills is an important factor in young people's decision to start—or not start—their own business, and it is relatively more important for younger people than older people.*

Regarding gender differences, it has been found that young German men have stronger entrepreneurial intentions than their female counterparts. While 19% of young men considered entrepreneurship an attractive avenue or career, only 12% of women felt the same. Moreover, only 53% of young women believed that they have entrepreneurial abilities compared to 62% of men. Interestingly, a person's level of education had a significant impact on these answers as well, as most of those who considered themselves to have the necessary skills for entrepreneurship have completed their Abitur. This aligns with prior theory, as several studies have shown that women are more likely to suffer negative consequences from pursuing entrepreneurial intentions than men (Vamvaka et al. 2020). We argue that the lower start-up rates among women in general, and in Germany, are neither innate nor instilled in childhood or early adolescence but are only justified in later phases of life and career, as men and women *then* encounter different framework conditions for the implementation of their entrepreneurial intentions.

**Hypothesis 2.** *Young women do not differ from young men in terms of entrepreneurial propensity and the obligatory person- and environment-related factors, but these gender differences become relevant for people over 25 years of age.*

The fear of failure as a reason for not realizing an existing start-up intention is dependent on experience, and it is a frequently studied term in entrepreneurship literature (see Section 2). If a person has already had the experience of being discriminated against socially and/or among friends or family in the event of failure in their own new business (or has lost creditworthiness with banks), they are less likely to consider starting another business in the future. In addition, in Germany—in contrast to the US, for example—a second chance is rarely granted after an initial failure. As this is widely known in German society and does not only apply to start-ups, non-founders also know this. Ergo, fear of failure is likely to have a negative effect on the start-up decision for all (young and old)

people in Germany. This effect, according to our hypothesis, is smaller for younger people than for older people, as they have had fewer experiences with failure than older people.

**Hypothesis 3.** *Fear of failure prevents young people from starting a business, though less frequently than older people.*

2.3.2. External or Contextual Factors

Regarding the contextual factors in Germany, this paper considers the following four for its conceptual model:

- Knowing other entrepreneurs who recently started a new business;
- Start-up opportunities in the region in which the respondent is living (own perception);
- Starting a new business as a desirable career choice (own perception of other people's opinions);
- Level of status and respect for those successfully starting a new business (own perception of other people's opinions).

As demonstrated in Section 2.1, the literature also discusses standard external factors of entrepreneurship such as investment opportunities, profit, reputation and media coverage of new businesses (Tubadji et al. 2021), as well as societal and cultural factors (Geldhof et al. 2014; Maleki et al. 2021; Tubadji et al. 2021). Upon analysing the societal factors, Pilkova et al. (2019) found that in developed countries such as Germany, the youth might be discouraged from entrepreneurial actions, as they have a wide range of appropriate job opportunities, which could make entrepreneurship a less attractive career option for them. Lastly, taking societal and cultural factors into consideration, research has proven that a nation's supportive culture and general public support can positively influence one's entrepreneurial intentions. It has also been found that cultural aspirations for excellence and competitiveness have positive influenced youth entrepreneurship (Lee et al. 2005). Younger people, all other things being equal, have naturally had less personal experience and met fewer people in person than older people. This also applies to experience with start-ups and knowledge of other entrepreneurs. Therefore, other factors that are not dependent on the amount of personal experience are relatively more important for younger people than for older people when deciding for or against starting a business. This is especially true for personal factors, while contextual factors are relatively more important for older people, as life and work experience facilitate a realistic assessment of these contextual factors.

**Hypothesis 4.** *For young people, their own perception of the entrepreneurial environment factors has a weaker influence on their start-up decision than personal factors than for older people.*

**Hypothesis 5.** *For young people, knowing other entrepreneurs is a less significant factor in the start-up decision than for older people.*

Figure 1 shows our conceptual model with five hypotheses and the factors in youth entrepreneurship in Germany.

The aim of this paper is to test the five hypotheses mentioned above on the basis of a suitable dataset and by means of adequate statistical methods, and thus to contribute to the state of empirical research on youth entrepreneurship.

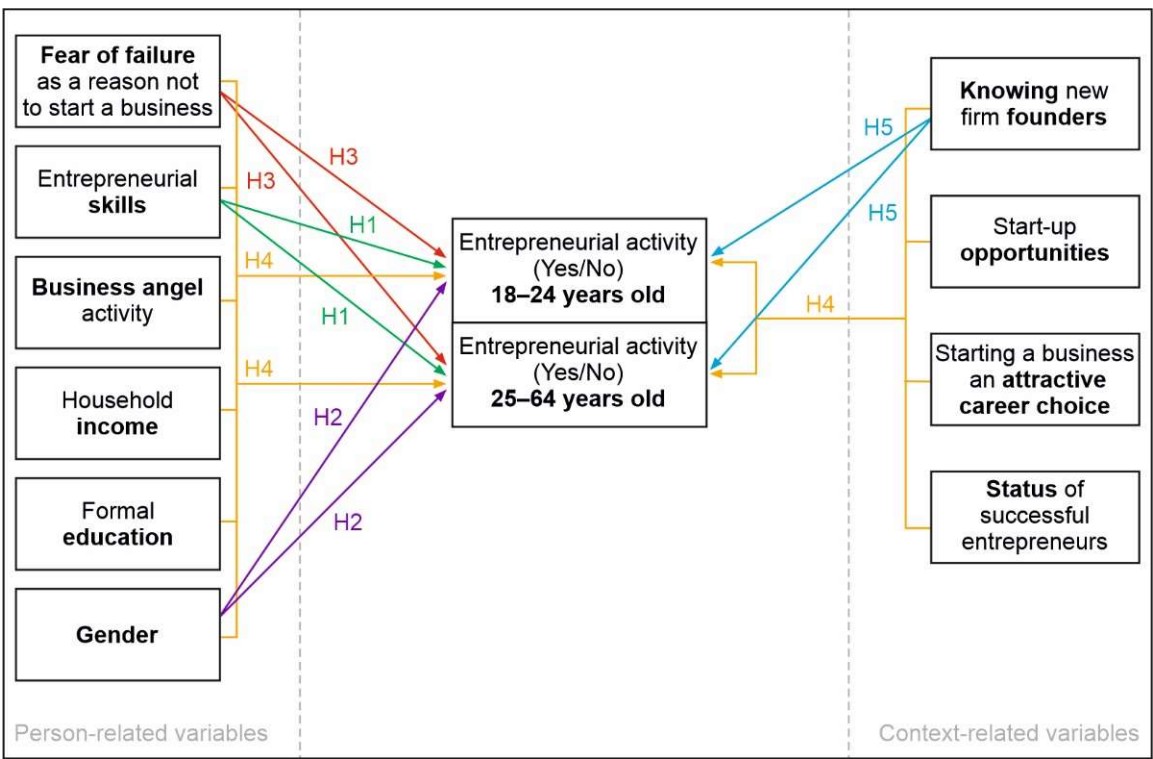

**Figure 1.** Conceptual model.

### 3. Data and Methods

*3.1. Data*

The empirical core of this paper is based exclusively on one data source, which, given the research questions of the paper, is without alternative from both a quantitative and qualitative perspective. We use the German data from the APS of the GEM. One of the many strengths of this entrepreneurship research consortium (the largest worldwide) is the fact that statistically representative data for entrepreneurs (i.e., founders of new businesses) and non-entrepreneurs are available for each country participating in the GEM in the respective survey year including the exact age of the person at the time of the survey. Empirical analyses of Germany, the subject of this paper, can also benefit from the fact that the German GEM team has been involved in this consortium since its inception and that, except for 2007, data are available for each year between 1999 and 2022 (see also the complete list of all annual GEM Country Reports Germany from 1999 to 2022: https://www.iwkg.uni-hannover.de/en/research/research-projects/details/projects/global-entrepreneurship-monitor-gem-country-report-germany, accessed on 22 March 2023; the most recent report also describes the core variables (Sternberg et al. 2022)). In principle, a very long data series with hundreds of thousands of cases is available for Germany from the GEM. In addition to the country reports, the global consortium publishes an annual global report (for the most recent one see GERA 2023). For details on GEM data and methods, see Reynolds et al. (2005) as well as the publicly available GEM Wiki (http://gem-consortium.ns-client.xyz/about/wiki, accessed on 12 June 2012). The entrepreneurship literature contains hundreds of empirical studies based exclusively or partially on GEM data, many of which focus on one country or compare several countries. The completely standardized data for a survey year for the countries involved in each case allow such comparisons to be made completely, or at least to a large extent in the case of intertemporal comparisons for the same or several countries.

Our paper uses data from 2009 to 2018 and pools the data from the individual 10 years to generate sufficiently large samples for the very rare event 'youth entrepreneurship'. (Please note that, in Germany, entrepreneurship in general is a rare event.) The APS is a

household survey (not a survey of entrepreneurs) in which—on average—only some 5% of respondents in Germany are founders of a new business according to the GEM definition. Thus, pooling many years (and controlling for them in the regression models) is crucial. Nevertheless, there is no other data source in Germany that offers more data on youth entrepreneurship over a longer period than the GEM data mentioned.

The target population of GEM is 18–64-year-olds. For our paper, we define 'youth' as 18–24-year-olds, although we could, in principle, also conduct analyses for individual age cohorts. We restrict our analysis to the 2009–2018 survey years. For some variables, analyses for the 2001–2022 period (with more than 105,000 cases for Germany) would also be possible; however, we will only use this for a descriptive overview of the development of the entrepreneurial activity rate over time. Due to changes in the scaling of variables and/or the wording of some variables used since 2019 or before 2009, we will only use data from the 2009–2018 period with 46,937 cases for all other analyses.

The data we use describe (e.g., demographic) characteristics of the individuals surveyed, the new businesses they may have founded and the respondents' perception of the context. Following our conceptual model, the nine independent variables are assigned to two groups: six internal factors describing certain personal attributes of the interviewee and four external factors describing the context in which the interviewee is living and working (in part, however, seen and perceived by the interviewee). All variables are dichotomous, except for household income (which we are using as a dummy variable in our regressions, too). See Table 1 for the definitions, descriptions, values and some measures of each of these independent variables.

**Table 1.** Definitions, descriptions and values of the variables.

| Variable Name | Variable Definition | Values |
|---|---|---|
| *Dependent* | | |
| TEA | Total Early stage Entrepreneurship: respondent involved in a nascent firm or young firm or both | 1: yes, 0: no |
| *Independent: person-related* | | |
| Businessangel | Respondent has, in the past three years, personally provided funds for a new business started by someone else, excluding any purchases of stocks or mutual funds | 1: yes, 0: no |
| Education | Respondent's highest educational attainment | 1: secondary degree and more 0: below secondary degree |
| Fearoffailure | Fear of failure would prevent the respondent to start a business | 1: yes, 0: no |
| Gender | Respondent gender | 1: female, 0: male |
| HHIncome | Household income (recoded into thirds) | 68,100: highest 33% tile 3467: middle 33% tile 33: lowest 33% tile |
| Skills | Respondent thinks to have the required knowledge/skills to start a business | 1: yes, 0: no |
| youngornotyoung | Respondent's age category | 1: 25–64, 0: 18–24 |

**Table 1.** *Cont.*

| Variable Name | Variable Definition | Values |
| --- | --- | --- |
| *Independent: context-related* | | |
| Goodopp | Respondent sees good opportunities for starting a business in the next 6 months in the area where he/she lives | 1: yes, 0: no |
| Career | In Germany, starting a business is considered a good career choice | 1: yes, 0: no |
| Respect | In Germany, people growing a successful new business receive high status | 1: yes, 0: no |
| Knowent | Respondent knows a person who started a business in the past 2 years | 1: yes, 0: no |

*3.2. Methods*

When interviewing the general adult population, involvement in early-stage entrepreneurial activities can be characterized in a binary manner. A person is either actively pursuing a venture creation, be it alone or as part of a group, or is not involved in any such process. The Global Entrepreneurship Monitor captures this information as 'Total early-stage Entrepreneurial Activity' (TEA) with 1 = Yes and 0 = No. Thereby, binary logistic regression models (see Equation (1)) are well suited to analyse the factors influencing new venture creation and are specified below:

$$\ln\left(\frac{P}{1-P}\right) = \beta_0 + \beta_1 X_1 + \beta_2 X_2 + \ldots + \beta_k X_k \tag{1}$$

The independent variables used in the models were described in Section 3.1. As Table A1 illustrates (see Appendix A), significant correlations exist between some independent variables; however, these are rather weak and should not be an issue for subsequent modelling.

Additionally, this paper focuses on youth entrepreneurship, which can be included in the models by either splitting the dataset into younger (18–24) and older (25–64) individuals and fitting models for each, or by including a binary indicator of a person's age group as an independent variable within an overall model. Therefore, a total of seven models, as described in Table 2, is included in this study. Please note that throughout the paper we refer to the models by their respective model number, as indicated in Table 2, i.e., the same model maintains the same number in all tables of Section 3.

Before analysing any final model, the underlying dataset contains a few particularities that need to be addressed. By pooling data from 10 years of data collection (2009–2018), the sample size was increased to 46,937 cases, facilitating a sufficient number of observations, especially among young entrepreneurs. However, since TEA is a rare event, with only 6.1% of the respondents being involved, the model may not be ideal for predictions on unknown data but rather for statistical inferences on the underlying dataset. Furthermore, due to the extended timeframe of 10 years, during which the data was collected, the year of data collection may have an effect, be it fixed or random. Although globally impactful events such as COVID-19 were not observed during these years, the data were evaluated for such effects regardless. First, to include the survey year as a fixed effect, a dichotomous indicator for each survey year was generated. Each model described in Table 2 was then fitted with and without these variables. Although indicators for single years occasionally became significant, subsequent analysis of variance (ANOVA), documented in Table A2 (see Appendix A), as well as a comparison of the goodness-of-fit as indicated in the McFadden's $R^2$ and the AIC did not reveal meaningful explanatory power in the

survey year. Considering this, it was deemed to be of little relevance as a fixed effect in the subsequent models.

**Table 2.** Model variants.

| Model Number | Included Independent Variables | Included Data |
|:---:|:---:|:---:|
| 1 | Youngornotyoung | All |
| 2 | Person-related variables + youngornotyoung | All |
| 3 | Person-related variables + context-related variables + youngornotyoung | All |
| 4 | Person-related variables + youngornotyoung | Only Young |
| 5 | Person-related variables + context-related variables + youngornotyoung | Only Young |
| 6 | Person-related variables + youngornotyoung | Only Not Young |
| 7 | Person-related variables + context-related variables + youngornotyoung | Only Not Young |

Second, a null two-level model with the survey year included as the Level 2 random effect was evaluated in comparison to a null single-level model without random effects, with little variance between the years being observed. Extending the full models by a Level 2 random effect (survey year) and calculating the intraclass correlation coefficients (ICC) as illustrated in Table A3 (see Appendix A) further confirms the lack of variance across years and thus does not warrant inclusion of random effects. Furthermore, when comparing the AIC, little improvements of mixed-effects models over fixed-effects models could be determined.

Third, each model was executed for each year individually, with the results being comparable and therefore robust with the model. Overall, it can be concluded that the survey year does not have a relevant influence on the model outcome.

To further evaluate the model results, each model was tested for multicollinearity (Table A4—see Appendix A) and goodness-of-fit (Hosmer–Lemeshow test; McFadden's $R^2$) as well as outliers in the deviance residuals and leverage. Multicollinearity was deemed to be a non-issue and assessed by means of the Generalized Variance Inflation Factor (GVIF) as presented by Fox and Monette (1992). Additionally, as illustrated in Table A5 (see Appendix A), the one-dimensional GVIF ($GVIF^{\frac{1}{(2*Df)}}$) does not exceed the threshold of $\frac{1}{1-R^2}$ (Vatcheva and Lee 2016) except on interaction terms, which is to be expected.

Outliers in deviance residuals remained acceptable, with the maximum peaking at 3.32 and the minimum at $-1.19$ at the tail ends. Outliers in leverage were compared to the model variant when excluded, and the influence was deemed low. Conclusively, no observations were excluded from the dataset in the final models. Furthermore, the models were compared to their respective null-models, with each being significantly better. Beyond fitting the models year-by-year, several measures to ensure robustness were undertaken. Given the rarity of entrepreneurship in Germany, the dependent variable is imbalanced. Therefore, the models may suffer from small sample bias, as described by King and Zeng (2001), although the imbalance is less prevalent in the dataset used for this study. To investigate this, a penalized likelihood logistic regression (Firth 1993; Heinze and Schemper 2002) was fitted for each of the models and compared to the unadjusted logistic regression outputs. Only miniscule deviations could be observed; therefore, small sample bias was deemed a non-issue. In addition, each model was evaluated against a probit model of identical specification, with the logit model performing slightly better, although with overall comparable results. Evaluating the models further, the rarity of entrepreneurship

impedes procedures such as K-Fold cross-validation. There are, however, ways, to reduce this imbalance to receive accurate results. Consequently, K-Fold cross-validation with 10 folds and 10 repetitions was performed, using the 'Synthetic Minority Oversampling Technique' (SMOTE) as well as 'Random Over-Sampling Examples' (ROSE). Both yielded similar accuracy measures for the models, as showcased in Table A6 (see Appendix A) and generally exhibited low variability. Additionally, Table A6 also provides the mean area under the ROC curve (AUC) after randomly splitting the dataset into training and testing data (80% training to 20% testing ratio) 100 times and applying the models. Dispersion remained low and the models of this paper achieve acceptable (0.7–0.8) to excellent (0.8–0.9) discrimination, with only Model 1 being inadequate (Hosmer et al. 2013).

Overall, the models do not exhibit any issues and provide a McFadden's $R^2$ of up to 0.2. While some residuals can certainly be considered outliers, their overall effect on the model is low and the median of the residuals is close to zero.

## 4. Empirical Results

### 4.1. Descriptive Results

Before proceeding with the multivariate analysis, Figures 2 and 3 provide first descriptive information on the TEA of young people in Germany. Figure 2 illustrates the development of the TEA rate of both younger and older people over 23 years 2001–2022 for which these data are available in Germany. In most years, and on average over the whole period, the TEA rate of 18–24-year-olds is lower than that of 25–64-year-olds. However, this is different in individual years, with obvious differences between the two age categories. It should be noted that over this relatively long period, both global and national contexts can change, which can influence start-up activities as well as age-cohort-specific TEA rates. For example, as Figure 2 shows, the TEA rate of younger people was relatively and absolutely high during the new economy boom at the beginning of this millennium. The global economic and financial crisis of 2008/2009 had a negative impact on the start-up rate in many countries, but not Germany (Hundt and Sternberg 2014). The COVID-19 pandemic (from 2020 to 2022) had a profound impact on TEA rates in Germany (and other high-income countries): in the first year of the pandemic, after rising steadily in the years before the pandemic, TEA rates fell sharply, only to recover shortly afterwards and reach an all-time high for Germany in 2022 (Fritsch et al. 2021; Sternberg et al. 2022; GERA 2023).

As for our paper, important independent variables are only available for 2009–2018; thus, the multivariate analysis in our paper is limited to this period.

Figure 3 shows the mean TEA rates for our study period from 2006 to 2018, as well as the exact age cohorts (age at the time of the survey). Overall, the figure confirms the observations of many other countries, and Germany as well, that entrepreneurial activity across age shows an approximately inverted U-shaped course, i.e., the very young and very old cohorts have lower TEA rates than the middle-aged cohorts. A comparison with Figure 2 suggests that this inverse U-curve would be somewhat flatter if the entire period for which GEM data are available in Germany (2001–2022) was used. Especially at the beginning of this millennium and during the COVID-19 pandemic, the TEA rates of 18–24-year-olds in Germany were high in absolute and relative terms (compared to the TEA rates of 25–64-year-olds).

Figure 3 also shows that the 18–24 age group is relatively heterogeneous in terms of TEA rates. It ranges from 3.8% for 18-year-olds to just under 6.5% for 24-year-olds and increases with age (but not for each cohort), although not steadily. Within this context, a note on the statistical representativeness of the GEM data was used. In our sample, 18–24-year-olds account for 10.8% of all 18–64-year-olds interviewed (or 5060 of the 46,937 persons interviewed). For the individual cohorts of 18–24-year-olds, there are between 662 cases (for the 20-year-olds, 1.4% of the total sample) and 766 cases (for the 24-year-olds, 1.9% of the total sample). These percentages are sufficiently close to the corresponding figures for Germany as a whole. According to the German Statistical Office (Statistisches Bundesamt [German Statistical Office] 2023), in the year 2018, a total of 10.4% of 18–64-year-olds

were between 18 and 24 years old. This value deviates only slightly from the above-mentioned reference value of our sample (10.8%). Even for the seven individual cohorts of 18–24-year-olds, the differences between the sample and population are never higher than 0.12 percentage points.

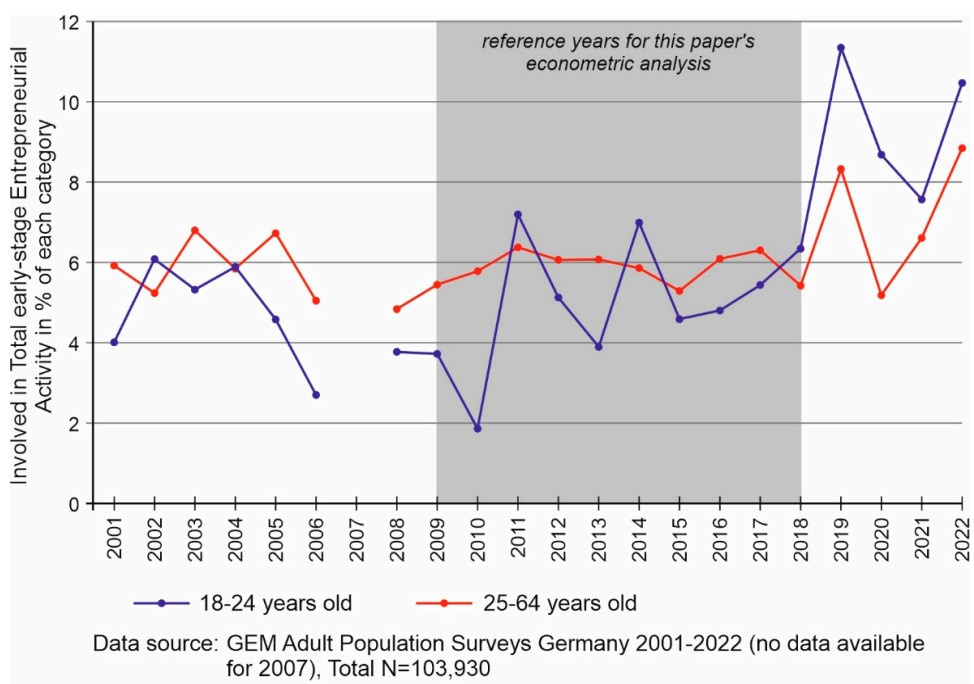

**Figure 2.** Annual TEA rates by age categories in Germany 2011–2022.

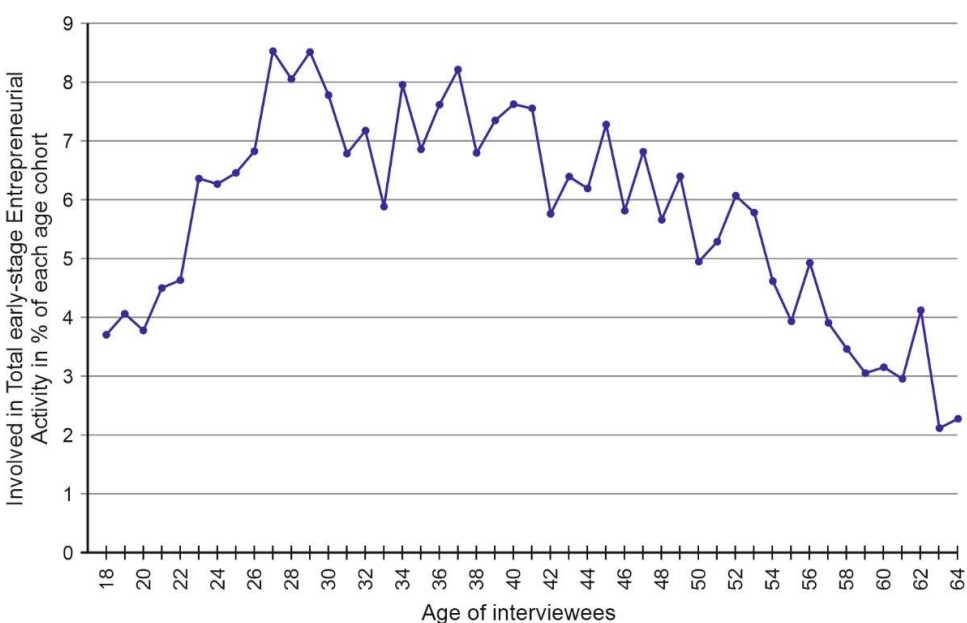

**Figure 3.** TEA rates by age categories in Germany 2009–2018.

Before we deal with the test of the five hypotheses using logistic regression models in the next chapter, we present another important descriptive result in Table 3, namely the comparison of founders and non-founders, each differentiated according to the two age categories. It is quite clear that the young founders are much more often male, have a lower

household income and are less convinced of their entrepreneurial skills than the older founders. In contrast, there are no statistically significant differences between younger and older founders in terms of the fear of failure as a barrier to founding, the level of formal education and business angel activity. Both subpopulations differ significantly in two of the four contextual factors: young founders recognize more respect for successful founders in German society and view the start of their own business as an attractive career option compared to the older founders. When comparing young founders with young non-founders, the expected higher values for founders emerge, for example, in the assessment of founding skills and entrepreneurial opportunities in the knowledge of other entrepreneurs and household income. The significantly higher proportion of men among the founders is also to be expected. In contrast, both subgroups of young men do not differ in their assessment of the status and respect of founders in society. The significantly more frequent significant coefficients in the group of non-founders, on the other hand, should not be overinterpreted, as they are at least partially due to the significantly larger number of cases compared to the founders.

**Table 3.** Characteristics of founders and non-founders compared by age.

| Founders | Mean Young | Mean Not Young | Pearson's $\chi^2$ | Sig |
|---|---|---|---|---|
| *Person-related factors* | | | | |
| Businessangel | 0.10 | 0.10 | 0.00 | |
| Education | 0.92 | 0.91 | 0.20 | |
| Fearoffailure | 0.17 | 0.20 | 1.50 | |
| Gender | 0.24 | 0.38 | 16.15 | *** |
| HHIncome (annual, 1000 €) | 26,618.23 | 31,611.57 | 15.51 | *** |
| Skills | 0.67 | 0.87 | 61.97 | *** |
| *Context-related factors* | | | | |
| Career | 0.63 | 0.44 | 30.84 | *** |
| Goodopp | 0.58 | 0.63 | 1.67 | |
| Knowent | 0.62 | 0.66 | 1.45 | |
| Respect | 0.79 | 0.71 | 6.35 | ** |
| **Non-founders** | | | | |
| *Person-related factors* | | | | |
| Businessangel | 0.03 | 0.04 | 16.58 | *** |
| Education | 0.89 | 0.83 | 102.96 | *** |
| Fearoffailure | 0.39 | 0.47 | 91.91 | *** |
| Gender | 0.45 | 0.50 | 38.38 | *** |
| HHIncome (annual, 1000 €, categories) | 18,372.34 | 25,195.37 | 406.31 | *** |
| Skills | 0.20 | 0.44 | 975.48 | *** |
| *Context-related factors* | | | | |
| Career | 0.64 | 0.47 | 417.67 | *** |
| Goodopp | 0.40 | 0.38 | 8.02 | *** |
| Knowent | 0.27 | 0.26 | 2.00 | |
| Respect | 0.84 | 0.78 | 86.47 | *** |

Note: ** $p < 0.05$, *** $p < 0.01$.

### 4.2. Regression Models

Table 4 illustrates the results of the first three models with *all* observations, regardless of age included in the underlying dataset. The odds ratios are mostly in line with the hypothesized outcomes. However, some reveal striking implications. Fear of failure seems to be a major deterrent to starting a business, while confidence in the skills and knowledge of an entrepreneur are major drivers of TEA. Remarkably, household income has a negative effect if it is larger than two-thirds of the population, but it shows no impact otherwise. Additionally, the lower likelihood of starting a business as an older entrepreneur is revealed. However, when this interacted with gender, it can be observed that women are more likely to start a business at an older age, while men usually do so earlier in life. In any case, age plays an important role in the probability of founding a company, which subsequently legitimizes several of our hypotheses. This also applies to the strong influence of skills, even independent of the age of the respondents. The $R^2$ increases noticeably if the contextual factors are also taken into account in Model 3 in addition to the person-related factors. When comparing Models 2 and 3, most of the odds ratios remain stable and statistically significant after adding the four contextual factors (three of which are themselves significant).

**Table 4.** Factors affecting the decision to start a business in total population.

| Independent Variables | Model 1 Odds Ratios | Model 2 Odds Ratios | Model 3 Odds Ratios |
|---|---|---|---|
| *Person-related factors* | | | |
| Businessangel (1 = yes, 0 = no) | | 1.600 *** (0.08) | 1.104 (0.086) |
| Education (1 = yes, 0 = no) | | 1.632 *** (0.082) | 1.440 *** (0.091) |
| Fearoffailure (1 = yes, 0 = no) | | 0.400 *** (0.056) | 0.455 *** (0.063) |
| Gender (1 = female, 0 = male) | | 0.542 *** (0.18) | 0.584 *** (0.193) |
| Youngornotyoung (1 = not young, 0 = young) | 1.241 *** (0.069) | 0.588 *** (0.097) | 0.593 *** (0.105) |
| Youngornotyoung # Gender (not young.female) | | 1.557 ** (0.186) | 1.502 ** (0.201) |
| HHIncome (lowest 33% tile) (1 = yes, 0 = no) | | 0.947 (0.063) | 0.977 (0.07) |
| HHIncome (highest 33% tile) (1 = yes, 0 = no) | | 0.977 (0.054) | 0.844 *** (0.061) |
| Skills (1 = yes, 0 = no) | | 6.268 *** (0.062) | 4.468 *** (0.069) |
| *Context-related factors* | | | |
| Career (1 = yes, 0 = no) | | | 0.993 (0.053) |
| Goodopp (1 = yes, 0 = no) | | | 1.64 *** (0.054) |
| Knowent (1 = yes, 0 = no) | | | 3.481 *** (0.055) |
| Respect (1 = yes, 0 = no) | | | 0.793 *** (0.06) |
| Intercept | 0.05 *** (0.066) | 0.031 *** (0.130) | 0.026 *** (0.156) |
| Goodness-of-fit | | | |
| $R^2$ | 0.00 | 0.13 | 0.18 |
| Hosmer-Lemeshow Sig. | - | 0.37 | 0.77 |
| Observations | 46,724 | 33,809 | 24,609 |

Note: ** $p < 0.05$, *** $p < 0.01$. Standard errors in parentheses. # means interaction term.

In the second set of models (Table 5), the sample was divided into 'young' (Models 4 and 5) and 'not young' individuals (Models 6 and 7), and Models 2 and 3 were applied again. Generally, the results of Model 2 and 3 can be confirmed; however, some nuances become visible. For one, formal education seems to be of little relevance in younger entrepreneurs but strongly significant in older founders. Opportunity recognition is more relevant for older entrepreneurs, while having a higher income remains a deterrent.

**Table 5.** Factors affecting the decision to start a business among 18–24 aged vs. 25–64 aged.

| Independent Variables | Model 4 Odds Ratios | Model 5 Odds Ratios | Model 6 Odds Ratios | Model 7 Odds Ratios |
|---|---|---|---|---|
| *Person related factors* | | | | |
| Businessangel (1 = yes, 0 = no) | 1.529 (0.275) | 1.116 (0.287) | 1.605 *** (0.084) | 1.099 (0.091) |
| Education (1 = yes, 0 = no) | 1.284 (0.291) | 1.100 (0.304) | 1.673 *** (0.085) | 1.478 *** (0.096) |
| Fearoffailure (1 = yes, 0 = no) | 0.341 *** (0.207) | 0.355 *** (0.219) | 0.403 *** (0.059) | 0.465 *** (0.066) |
| Gender (1 = female, 0 = male) | 0.585 *** (0.184) | 0.625 ** (0.198) | 0.839 *** (0.051) | 0.873 ** (0.057) |
| HHIncome (lowest 33% tile) (1 = yes, 0 = no) | 0.837 (0.199) | 0.922 (0.214) | 0.969 (0.066) | 0.996 (0.075) |
| HHIncome (highest 33% tile) (1 = yes, 0 = no) | 1.283 (0.199) | 1.220 (0.214) | 0.959 (0.056) | 0.819 *** (0.064) |
| Skills (1 = yes, 0 = no) | 7.076 *** (0.164) | 5.373 *** (0.175) | 6.156 *** (0.067) | 4.357 *** (0.075) |
| *Context related factors* | | | | |
| Career (1 = yes, 0 = no) | | 1.057 (0.177) | | 0.985 (0.056) |
| Goodopp (1 = yes, 0 = no) | | 1.207 (0.171) | | 1.704 *** (0.057) |
| Knowent (1 = yes, 0 = no) | | 2.944 *** (0.177) | | 3.539 *** (0.058) |
| Respect (1 = yes, 0 = no) | | 0.931 (0.22) | | 0.782 *** (0.062) |
| Intercept | 0.035 *** (0.333) | 0.031 *** (0.414) | 0.018 *** (0.106) | 0.015 *** (0.133) |
| Goodness-of-fit | | | | |
| $R^2$ | 0.17 | 0.20 | 0.12 | 0.18 |
| Hosmer-Lemeshow Sig. | 0.99 | 0.92 | 0.25 | 0.93 |
| Observations | 3086 | 2303 | 30,723 | 22,306 |

Note: * $p < 0.1$, ** $p < 0.05$, *** $p < 0.01$. Standard errors in parentheses.

Table 5 provides all the results needed to decide on our five hypotheses. None of the 10 independent variables examined in the models has a stronger influence on the TEA variable than the perception of one's own entrepreneurial skills. This is true for each of the Models 4–7, i.e., for both younger and older people. The probability that the respondent will start a business is between four and six times higher for a person who believes that they have the necessary skills to start a business than for those who do not. The odds ratios are even higher for younger people than for older people. Comparing the models (Models 5 and 7) with and without the contextual variables (Models 4 and 6), the odds ratios are only slightly reduced for the former. Overall, this results in a very convincing confirmation of Hypothesis 1.

In all models, women show a lower probability of starting a business than men, regardless of age. However, in each of the four models (as well as Models 2 and 3 in Table 4), the statistically highly significant odds ratios for older women are less far removed from those of (older) men than is the case for younger women. Thus, women start businesses less often and later in life than men. Hypothesis 2 must therefore be rejected.

If a respondent might not start a business because of the fear of failure, this reduces the actual probability of starting a business by approximately 65% for younger people and by only 55% for older people, compared to people who do not have this fear of failure. These odds ratios are also statistically significant in all models. Due to the proven (although not

very significant) differences between the impact on younger and older people (stronger for the former), we consider Hypothesis 3 to be mainly not confirmed.

For young people, the four variables regarding their own perception of the entrepreneurial environment factors are not statistically significant in three out of four cases, and only the odds ratios of knowing another entrepreneur are high and significant. Among the older respondents, on the other hand, the odds ratios of these contextual factors are significant in most cases. The positive evaluation of entrepreneurial opportunities and the acquaintance of another entrepreneur are the variables with the 3rd and 2nd strongest influence of all 10 independent variables among older people. We can therefore consider Hypothesis 4 to be confirmed.

For both older and younger people, knowing another entrepreneur (who has founded a business in the last two years) significantly increases their probability of founding a business. It is 2.9 times higher for younger people and 3.5 times higher for older people than for people who do not know such entrepreneurs personally. The difference between the two age categories is noticeable, although not very large. We can thus essentially confirm Hypothesis 5.

In addition to these results concerning the five hypotheses, the regression models show another interesting finding. For older people, a high level of formal education increases the probability of founding a business by a factor of 1.5 to 1.7 (compared to people with a lower level of education), while no statistically significant influence can be found for younger people.

## 5. Discussion of Results

Our empirical findings documented in Chapter 4 show two results. First, young people differ considerably from older people in terms of the likelihood of starting a business and, in particular, the factors that explain this likelihood. While the lower likelihood of younger people compared to older people (especially the middle-aged cohorts), which we also identified, has been shown before, at least for most high-income countries (e.g., Schøtt et al. 2015), this paper makes an important contribution to the explanatory factors of these differences, differentiated according to person-related and contextual factors. This is not found in previous literature and data for an entire country. To the best of our knowledge, however, this paper is the first to examine these factors by differentiating between younger and older people in Germany. Second, the empirical findings show that young founders also differ considerably from young non-founders in terms of important demographic characteristics and the assessment of the entrepreneurial context. For most of our 10 independent variables, these intragenerational differences are similar to those of older people (especially for gender and entrepreneurial skills). Both core results justify a specific consideration of young people in the analysis of start-up frequencies and their factors for selected countries or subnational regions. In other words, one should not assume that an analysis of these factors for a country and all adults will adequately reflect start-up behaviour and its factors for younger people.

Of the five hypotheses tested, three could be confirmed without limitations and two could not. Overall, this attests to an appropriate and open-ended analysis strategy, albeit with limitations, as discussed in Section 5. Our analysis of Hypothesis 1 shows that the positive perception of one's own entrepreneurial skills is not only a significant factor in young people's decision to start a new business but also the one with the highest odds ratios of all 10 independent variables in all relevant regression models. The odds ratios for this variable are all positive and statistically significant, and they show that young people with a positive self-perception are between five and seven times more likely to start a business than those who do not have this positive perception. This confirms the findings of Schøtt et al. (2015) and Hekman (2007), even if they do not explicitly refer to young people in Germany or to the aforementioned self-efficacy in entrepreneurial skills. However, the sources mentioned do not examine the impact of this self-efficacy on the actual decision to start a business. To our knowledge, our analysis is the first to illustrate the significance of

the perception of one's own entrepreneurial skills explicitly for young people in Germany and in relative terms compared to older people.

Hypothesis 2 is not supported by our models. Contrary to expectations, the start-up frequency and factors in men already differ from women at a young age and not only when they are older. In fact, women do not completely catch up with men in terms of start-up activities at an older age, but they do at least partially. There must therefore already be person-related and/or contextual factors at a young age that cause young women to start businesses much less often than men. Our interaction term of the dummy variable age category (young–old) and gender shows a 1.5 times higher probability for women between the ages of 25 and 64 than for other people. This confirms recent—purely descriptive—statements of the GEM Women Report, according to which the ratio of TEA start-ups by women compared to men in Germany in 2021 is about 0.8 for 18–24-year-olds but 1.2 and 1.1 for 24–54-year-olds and 55–64-year-olds, respectively (GEM 2022). If the gender gap in start-ups is to be reduced, it is therefore necessary—also in Germany—to start doing so in the first two to three decades of young people's lives.

Our core argument in connection with the variable 'fear of failure as a reason not to start a business' was life and work experience, of which younger people naturally have less than older people. Younger people have not yet had as many negative experiences of failure and therefore cannot have had as many (negative) experiences with social consequences of individual failure as older people, which is, we assumed, why they are less likely to refrain from starting a business, that is, because they fear the social or other consequences of a failed start-up (but for other reasons). The result is therefore quite surprising, as many studies show that because older people can lose more on average through a risky start-up than younger people, risk aversion is therefore higher among older people (Schøtt et al. 2015; Mather et al. 2012). Apparently, there are other factors (not covered in the canon of our independent variables) associated with the fear of failure variable that explain why younger people are more influenced by the fear of failure than older people. Conceivable factors are a fundamentally lower self-efficacy (which we actually observed in relation to entrepreneurial skills in Hypothesis 1) or a fundamentally more pessimistic view of the future among younger people (i.e., those who were younger earlier) in view of many medium- or long-term crises (pandemic, climate change, etc.).

For young people, the contextual factors studied have less influence on the start-up decision than personal factors than for older people (Hypothesis 4). For some contextual factors, this is more than plausible and confirms older findings. For example, the two person-related factors of gender and self-efficacy in entrepreneurial skills have a significant influence on the start-up decision in general, but this influence is stronger for younger people than for older people. The network and role model variable investigated in Hypothesis 4 is the only one with a strong effect on the start-up decision among younger people, unlike among older people, for whom three of the four contextual variables exert a strong influence. We interpret this in the sense that younger people, due to less professional and life experience, know and assess their own person (relatively) better than the (spatial and other) context. For older people, on the other hand, it tends to be the other way round. This greater ability to assess personal factors leads to the latter being given greater weight in such an important decision in favour of or against an entrepreneurial career than is the case with older people. In our opinion, this comparison of a total of 10 factors in the same regression models and with concrete reference to a country—and on an empirically sufficient basis—is one of the important contributions that our paper makes to the literature. For young people, the context (here especially related to the regional context) is less important than for older people. The widespread postulate in regional entrepreneurship research that the regional environment is an important cause of the start-up decision and subsequent start-up success (e.g., Dahl and Sorenson 2012) may therefore apply more to older than younger entrepreneurs.

As the analysis of Hypothesis 5 has shown, the acquaintance of another entrepreneur has a strong and positive influence on the probability of founding a company. This influence is present for both younger and older people, but it is even stronger for the latter, as expected. We interpret this finding in the light of the role model argument (Fritsch and Wyrwich 2018; Wyrwich et al. 2019). According to this, there is both a positive and a negative side to role models. If the entrepreneur with whom our respondents are acquainted has had good experiences with their start-up (e.g., founded very successfully and/or experienced great satisfaction in this type of gainful employment), then this could have a positive effect on their acquaintance and entrepreneurial intention ('if they can do it, I can, too' (Sorenson and Audia 2000, p. 443). Conversely, negative experiences can have the opposite effect on the acquaintance ('if they can fail, I can, too'). It is possible that younger people have more frequent contact with successful entrepreneurs than failed entrepreneurs—or that their perception of (successful and failed) founders in the media they use (possibly different from those used by older people) conveys a more positive image of entrepreneurs (Tubadji et al. 2021). In any case, the analysis shows for both age categories a clear confirmation of the impact of role models and network relationships with actual founders on the decision to found a company, which has previously been studied for Germany (see Brüderl and Preisendörfer (1998) on the network hypothesis for new firm founders in general) but without reference to age.

## 6. Implications for Policy and Further Research

The core results of the paper can be summarized as follows. First, young people differ considerably from older people in terms of the likelihood of starting a business and, in particular, the factors that explain this likelihood. Second, young founders differ considerably from young non-founders in terms of demographic characteristics (gender, household income, entrepreneurial skills) and the assessment of the entrepreneurial context (e.g., perception of entrepreneurial opportunities). For most of our 10 independent variables, these intragenerational differences are similar to those of older people (especially for gender and entrepreneurial skills). Third, the positive perception of one's own entrepreneurial skills is not only a significant factor in young people's decision to start a business, but also the one with the highest odds ratios of all ten independent variables in all relevant regression models. Fourth, men's likelihood of starting a business already differs from women at a young age. In fact, women do not completely catch up with men in terms of start-up activities at an older age, but they do catch up at least partially. Fifth, young people are *not* less likely than older people to refrain from starting a business because they fear the social or other consequences of a failed start-up (but for other reasons). Sixth, for young people, the contextual factors studied have less influence on the decision to start a business than the person-related factors as compared to older people. Finally, the acquaintance of another entrepreneur has a strong and positive influence on the probability of founding a company. This influence is present for both younger and older people, but it is even stronger for the latter.

These empirical results for Germany, the focus of the paper, give rise to several implications for start-up policy. If the goal of government policy is to generate the (mostly positive) economic effects of a high national entrepreneurship rate (Fritsch 2013), the relatively low start-up rate in Germany must increase. It is rational to start politically with those population groups whose start-up rates have been comparatively low so far, such as those of younger people. Even if not all of these policies generate the desired economic effects (Shane 2009) and only relatively few public policy programmes to assist new youth businesses are evaluated by independent and neutral researchers (cf. OECD 2023), such tax-funded policies are comparatively inexpensive and more sustainable attempts to support economic development and, in particular, regional structural change through the regular renewal of the business stock compared to efforts to promote large, old and established companies in Germany or even abroad. However, the great heterogeneity of new businesses must explicitly be taken into account (Minniti 2008).

The empirical results of the paper provide evidence for such entrepreneurship support policies in favour of younger people (Mariani et al. 2019). We argue that such policies should, if possible, explicitly address the empirically proven factors involved in youth entrepreneurship; in other words, no 'one size fits all' policy for new businesses, regardless of the age of the founders. Policymakers still too often develop programmes that are intended to appeal to all founders—and thereby unintentionally disadvantage the much more numerous older founders of new businesses. To develop good start-up policies for young people, one must know their motives and framework conditions and then develop suitable policy instruments. The various 'inclusive entrepreneurship' reports of recent years for Germany (OECD and European Commission 2021) show that, despite some efforts, there remains room for improvement, also in comparison with other countries. For younger people (defined in the paper as 18–24-year-olds) in Germany, self-efficacy in entrepreneurial skills, personal acquaintance of other entrepreneurs and fear of the social consequences of failure in a new business is particularly important when deciding to start their own businesses or not. In contrast, the formal educational status and the reputation of successful founders in society do not play an important role, which is quite different from older people. Policy measures in favour of new businesses of younger people should therefore aim to improve the entrepreneurial skills of this target group, which is naturally easier and earlier within the context of school education. However, for many years, and despite several initiatives, this has been a comparative weakness of the entrepreneurial context in Germany compared to other countries, as the annual GEM Country Report Germany has documented for many years (for the most recent one, see Sternberg et al. (2022)). The fear of failure as a start-up barrier—as a social phenomenon—is less easy (or even quick) to influence through start-up policy measures, while the personal acquaintance of other founders is not at all.

Of course, this paper also has some limitations. With a total of 10 independent variables, we have attempted to take into account both important person-specific and contextual factors. However, factors such as migration background or place of residence within Germany had to be omitted. This is especially true for some contextual factors (e.g., economic framework conditions). The number of cases, although higher in the GEM data set for Germany than any other country (apart from the UK and Spain) and other surveys worldwide, would have had to be even higher in individual years to be able to adequately conduct the very detailed logit regressions that require large samples. Therefore, we had to reduce the data analyses planned for the period 2001–2022, with more than 100,000 cases, to 2009–2018 (with a good 46,000 cases). Finally, it should be noted that our entire analysis focuses on young people in Germany as a whole, although interregional differences within Germany (as in most countries) in the start-up behaviour and attitudes of young men and women are likely (for example, between West and East Germany or between urban and rural regions).

The paper and the data used in it offer numerous opportunities for further research. In part, the corresponding topics can also be covered with (already existing) GEM data, but we have gone beyond the scope of this paper. This applies, for example, to the distinction between positive and negative role model effects or the influence of the pandemic (or other external shocks) on the self-efficacy for entrepreneurial skills—and thus on new businesses—among young people. The interregional differences in young people's entrepreneurial behaviour mentioned under limitations would also be worth including in the analysis from the perspective of young people in the future. The data from the annual GEM Adult Population Survey would also allow this in principle.

Data from sources other than GEM could enrich the analysis, especially for contextual factors (e.g., by adding economic or social characteristics of the region in which the person lives). Not only the start-up decision itself but also the factors of the subsequent success of the new business would be worth considering. They are potentially even more important in the economic effects desired by start-up policy than the decision to start a new business. The GEM also offers some data for these success characteristics (e.g., on the expected employment or turnover effects and export intensity).

**Author Contributions:** Conceptualization, R.S.; methodology, R.S.; software, R.S.; validation, R.S.; formal analysis, R.S.; investigation, D.B.; resources, R.S.; data curation, R.S.; writing—original draft preparation, D.B.; writing—review and editing, D.B.; visualization, R.S.; supervision, R.S.; project administration, R.S. All authors have read and agreed to the published version of the manuscript.

**Funding:** This research received no external funding.

**Informed Consent Statement:** Not applicable.

**Data Availability Statement:** All data stems from the publicly available database of the Global Entrepreneurship Monitor (GEM) website: gemconsortium.org (accessed on 1 March 2023).

**Conflicts of Interest:** The authors declare no conflict of interest.

## Abbreviations

| | |
|---|---|
| ANOVA | Analysis Of Variance |
| APS | Adult Populations Survey |
| GEM | Global Entrepreneurship Monitor |
| GVIF | Generalized Variance Inflation Factor |
| ICC | Intraclass Correlation Coefficients |
| OECD | The Organization for Economic Cooperation and Development |
| ROSE | Random Over-Sampling Examples |
| SMOTE | Synthetic Minority Oversampling Technique |
| TEA | Total early-stage Entrepreneurial Activity |
| UN | United Nations |

## Appendix A

**Table A1.** Pearson correlation coefficients of independent variables.

| | Gender | HHIncome | Education | Busangel | Skills | Fearoffailure | Goodopp | Career | Respect |
|---|---|---|---|---|---|---|---|---|---|
| Gender | | | | | | | | | |
| HHIncome | −0.09 *** | | | | | | | | |
| Education | 0.05 *** | 0.21 *** | | | | | | | |
| usangel | −0.06 *** | 0.09 *** | 0.04 *** | | | | | | |
| Skills | −0.16 *** | 0.17 *** | 0.08 *** | 0.10 *** | | | | | |
| Fearoffailure | 0.13 *** | −0.08 *** | −0.03 *** | −0.06 *** | −0.19 *** | | | | |
| Goodopp | −0.09 *** | 0.17 *** | 0.12 *** | 0.07 *** | 0.13 *** | −0.12 *** | | | |
| Career | −0.01 | −0.08 *** | −0.08 *** | −0.03 *** | −0.05 *** | −0.01 *** | 0.03 *** | | |
| Respect | 0.04 *** | 0.04 *** | 0.05 *** | −0.02 *** | −0.08 *** | 0.06 *** | 0.07 *** | 0.17 *** | |
| Knowent | −0.08 *** | 0.12 *** | 0.09 *** | 0.16 *** | 0.22 *** | −0.09 *** | 0.17 *** | −0.01 * | −0.03 *** |

Note: * $p < 0.1$, *** $p < 0.01$.

**Table A2.** Deviance of survey year and its significance in ANOVA.

| Year | Model 1 | Model 2 | Model 3 | Model 4 | Model 5 | Model 6 | Model 7 |
|------|---------|---------|---------|---------|---------|---------|---------|
| 2010 | 0 | 15 *** | 73 *** | 5 ** | 13 *** | 12 *** | 61 *** |
| 2011 | 5 ** | 1 | 19 *** | 1 | 5 ** | 1 | 16 *** |
| 2012 | 3 | 2 | 6 ** | 1 | 3 * | 2 | 4 * |
| 2013 | 1 | 0 | 7 *** | 0 | 0 | 0 | 8 *** |
| 2014 | 1 | 0 | 0 | 8 *** | 5 ** | 0 | 2 |
| 2015 | 1 | 1 | 3 * | 1 | 2 | 0 | 2 |
| 2016 | 0 | 1 | 0 | 0 | 0 | 1 | 0 |
| 2017 | 2 | 0 | 5 ** | 0 | 1 | 0 | 5 ** |
| 2018 | 1 | 5 ** | 17 *** | 0 | 1 | 5 ** | 17 *** |

Note: * $p < 0.1$, ** $p < 0.05$, *** $p < 0.01$.

**Table A3.** Summary of intraclass correlation coefficients (ICC).

| Model Number | ICC |
|--------------|-----|
| 1 | 0 |
| 2 | 0.003 |
| 3 | 0.028 |
| 4 | 0.011 |
| 5 | 0.045 |
| 6 | 0.002 |
| 7 | 0.027 |

**Table A4.** One-dimensional generalized variance inflation factors and $R^2$ threshold.

| Variable | Model 2 | Model 3 | Model 4 | Model 5 | Model 6 | Model 7 |
|----------|---------|---------|---------|---------|---------|---------|
| Businessangel | 1.01 | 1.016 | 1.015 | 1.025 | 1.009 | 1.015 |
| Education | 1.019 | 1.022 | 1.011 | 1.017 | 1.02 | 1.022 |
| Fearoffailure | 1.012 | 1.019 | 1.009 | 1.013 | 1.013 | 1.022 |
| Gender | 3.756 | 3.59 | 1.02 | 1.022 | 1.018 | 1.022 |
| HHIncome | 1.02 | 1.024 | 1.011 | 1.016 | 1.017 | 1.023 |
| Skills | 1.033 | 1.049 | 1.009 | 1.025 | 1.02 | 1.031 |
| Career | | 1.029 | | 1.013 | | 1.024 |
| Goodopp | | 1.027 | | 1.013 | | 1.03 |
| Knowent | | 1.03 | | 1.032 | | 1.031 |
| Respect | | 1.026 | | 1.018 | | 1.027 |
| youngornotyoung | 1.186 | 1.195 | | | | |
| youngornotyoung # Gender | 3.837 | 3.671 | | | | |
| $R^2$ Threshold | 1.145 | 1.215 | 1.205 | 1.251 | 1.14 | 1.212 |

# stands for interaction terms.

**Table A5.** Generalized variance inflation factors.

| Variable | Model 2 | Model 3 | Model 4 | Model 5 | Model 6 | Model 7 |
|---|---|---|---|---|---|---|
| Businessangel | 1.02 | 1.03 | 1.03 | 1.05 | 1.02 | 1.03 |
| Education | 1.04 | 1.04 | 1.02 | 1.03 | 1.04 | 1.05 |
| Fearoffailure | 1.02 | 1.04 | 1.02 | 1.03 | 1.03 | 1.04 |
| Gender | 14.11 | 12.89 | 1.04 | 1.04 | 1.04 | 1.04 |
| HHIncome | 1.08 | 1.1 | 1.05 | 1.07 | 1.07 | 1.09 |
| Skills | 1.07 | 1.1 | 1.02 | 1.05 | 1.04 | 1.06 |
| Career | | 1.06 | | 1.03 | | 1.05 |
| Goodopp | | 1.06 | | 1.03 | | 1.06 |
| Knowent | | 1.06 | | 1.06 | | 1.06 |
| Respect | | 1.05 | | 1.04 | | 1.05 |
| youngornotyoung | 1.41 | 1.43 | | | | |
| youngornotyoung # Gender | 14.72 | 13.48 | | | | |

# stands for interaction terms.

**Table A6.** Area under ROC curve and K-Fold Cross Validation Accuracy.

| Model | Measure | Mean | SD |
|---|---|---|---|
| Model 1 | AUC | 0.5107 | 0.0059 |
| Model 1 | K-Fold Cross Validation (SMOTE) | 0.1554 | 0.0045 |
| Model 1 | K-Fold Cross Validation (ROSE) | 0.1711 | 0.1109 |
| Model 2 | AUC | 0.7698 | 0.0096 |
| Model 2 | K-Fold Cross Validation (SMOTE) | 0.6724 | 0.0088 |
| Model 2 | K-Fold Cross Validation (ROSE) | 0.6707 | 0.0083 |
| Model 3 | AUC | 0.8109 | 0.0103 |
| Model 3 | K-Fold Cross Validation (SMOTE) | 0.7236 | 0.0099 |
| Model 3 | K-Fold Cross Validation (ROSE) | 0.7198 | 0.0096 |
| Model 4 | AUC | 0.7988 | 0.0358 |
| Model 4 | K-Fold Cross Validation (SMOTE) | 0.7968 | 0.0343 |
| Model 4 | K-Fold Cross Validation (ROSE) | 0.7854 | 0.0336 |
| Model 5 | AUC | 0.8094 | 0.0334 |
| Model 5 | K-Fold Cross Validation (SMOTE) | 0.7585 | 0.0254 |
| Model 5 | K-Fold Cross Validation (ROSE) | 0.752 | 0.0275 |
| Model 6 | AUC | 0.7662 | 0.009 |
| Model 6 | K-Fold Cross Validation (SMOTE) | 0.6591 | 0.0085 |
| Model 6 | K-Fold Cross Validation (ROSE) | 0.659 | 0.0107 |
| Model 7 | AUC | 0.8104 | 0.0101 |
| Model 7 | K-Fold Cross Validation (SMOTE) | 0.7247 | 0.0113 |
| Model 7 | K-Fold Cross Validation (ROSE) | 0.7172 | 0.0105 |

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
