# Peer review of "Youth Entrepreneurship in Germany: Empirical Evidence on the How, the Why, the How Many, the Who and the When"

_economies, doi:10.3390/economies11060161_

Round 1

Reviewer 1 Report

The problem of entrepreneurial activity occurs in all developed societies. Each of the societies struggles with the problem of how to encourage citizens to entrepreneurial activity, especially the younger part of society. Despite the fact that the problem is known, it is still insufficiently researched, which is why the article entitled "Youth Entrepreneurship in Germany: Empirical Evidence on the How, the Why, the How Many, the Who and the When" is a particularly valuable initiative aimed at getting to know the motivation for entrepreneurial activities.

Five hypotheses were formulated in the work, but the main goal (it remains in the sphere of guesswork) and specific goals that could be related to the hypotheses were not formulated. Therefore, I suggest that this shortcoming be rectified.

An appropriate amount of literature was cited in the reviewed work. The reviewer's reservations are also not raised by figures and tables.

The research procedures used in the work together with statistical methods are properly selected and also properly applied. The adopted research hypotheses (as many as five) were properly verified and the correct conclusions were drawn on their basis.

The whole work, despite the minor shortcomings indicated above, is written carefully and in accordance with the art of preparing scientific papers.

Author Response

Author's’ Response to Reviewer 1

Dear Reviewer 1,

many thanks for giving us the opportunity to resubmit a revised version of our paper entitled "Youth entrepreneurship in Germany: Empirical evidence on the how, the why, the how many, the who and the when submitted to “Economies”.

Your comments helped us to improve the previous version. In the remainder of this letter, we respond to your comments (in italics) that were put forward along with our responses.

Best regards,

The Authors

Comments to the Author

The problem of entrepreneurial activity occurs in all developed societies. Each of the societies struggles with the problem of how to encourage citizens to entrepreneurial activity, especially the younger part of society. Despite the fact that the problem is known, it is still insufficiently researched, which is why the article entitled "Youth Entrepreneurship in Germany: Empirical Evidence on the How, the Why, the How Many, the Who and the When" is a particularly valuable initiative aimed at getting to know the motivation for entrepreneurial activities.

Response:

Many thanks for your friendly comment at the beginning of your valuable review.

Comments to the Author

Five hypotheses were formulated in the work, but the main goal (it remains in the sphere of guesswork) and specific goals that could be related to the hypotheses were not formulated. Therefore, I suggest that this shortcoming be rectified.

Response:

Done. We have now integrated one sentence that defines the aim of our paper, related to the five hypotheses described before.

Comments to the Author

An appropriate amount of literature was cited in the reviewed work. The reviewer's reservations are also not raised by figures and tables.

Response:

Many thanks. Good to know that you acknowledge our efforts in citing and using the relevant literature on youth entrepreneurship and general and in Germany in particular.

Comments to the Author

The research procedures used in the work together with statistical methods are properly selected and also properly applied. The adopted research hypotheses (as many as five) were properly verified and the correct conclusions were drawn on their basis.

Response:

Many thanks, much appreciated.

Comments to the Author

The whole work, despite the minor shortcomings indicated above, is written carefully and in accordance with the art of preparing scientific papers.

Response:

Many thanks.

Reviewer 2 Report

Dear Authors, Please carefully check and solve all observations as formulated (below) and sent to the editors of the Economies MDPI journal to which you submitted your manuscript:

<<Dear Editors,

In this paper (id economies-2346607, entitled “Youth entrepreneurship in Germany: Empirical evidence on the how, the why, the how many, the who and the when”), the authors used statistically representative survey data from the Global Entrepreneurship Monitor for Germany and they applied logit regressions to determine the influence of ten independent variables on the likelihood to start a business. They claim that their results suggest that start-up promotion policies should explicitly address the empirically proven factors of youth entrepreneurship instead of a 'one size fits all' policy for new businesses regardless of the age of the founders.

After reading this paper, I think I found some issues the authors should deal with. Let’s start with the format ones and then continue with those related to the content:

  • The paper must follow all the instructions of the journal precisely indicated at: https://www.mdpi.com/journal/economies/instructions;
  • The chart in Figure 3 used the type dedicated to representing time series (e.g., Figure 2) although the variable on the X-axis is not represented by time or time stamps; In this particular case (Figure 3) the authors are trying to emphasize the relation between two variables indicated on those two axes. Therefore they should choose a scatter plot or similar, but not something dedicated to representing the evolution in time. Age is time-related but is not time;

  • All references to equations/formulas must be explicitly and precisely formulated in the main text For instance, “... (see eq. N).” but not “as follows” or similar;

  • The authors must avoid ending some sections/subsections with formulas, figures, tables, or other components (e.g., Fig. 1 just before section 3, Table 6 just before section 4, and no explanatory text after). The authors are required to check the entire manuscript for similar issues;

  • The authors must additionally ensure that all figures have the required resolution (min. 300 dpi, according to the Journal’s instructions at the link above). All three figures suffer in these terms;

  • There is a high number of tables. I counted 9 in this manuscript. Some of them which are considered by the authors not essential for understanding the main flow of ideas in the manuscript must be moved to the Appendix section. If this section is not existing, the authors must create one; 

  • The List of Abbreviations at the end of the manuscript is needed (I counted at least 6);

  • The Limitations should be a dedicated section;

  • There are no details about the AUC-ROC accuracy values of the logit models obtained. The authors should present them precisely for each logit regression model;

  • Each pair of predictors should be assessed in terms of collinearity: VIF against 1/(1-R^2) as indicated by Vatcheva et al.(2016) https://doi.org/10.4172/2161-1165.1000227&Freund et al.(2006) https://tinyurl.com/jypdexuc.

  • The authors are also required to include more explanations and precise details about the standard view of the accuracy values (>=70% and <80%-fair models; >=80% and <90%-good models; >=90%-very good/excellent models) and their application here. They should provide more references to scientific papers where this topic is considered and the accuracy intervals are precisely defined;

  • At first glance, the final list of references seems reasonable. Still, I counted one conference paper that is not considered finalized research work and should be replaced with a  reference to at least one journal paper;

  • Moreover, why discuss just the core results (section 5. Discussion of core results) and not all the results? The authors should precisely explain in the manuscript or rephrase;

  • The authors must understand that replicability as a fundamental principle in science (https://doi.org/10.1007/s10516-021-09610-2  https://doi.org/10.1038/nature.2016.20504 ) starts with data and it is not a fad but a necessity. Therefore, they should insert in the Data Availability Statement section at the end of the manuscript all precise links to all data providers’ / own datasets used when testing the entire approach;

  • Following this scientific principle above, the authors should provide full details about the software and hardware they used for collecting, transforming, analyzing the data, and reporting the results presented in this manuscript (all data transformation/treatment and regression script sequences required if used);

  • The authors must also triangulate using many methods and techniques (https://doi.org/10.1038/d41586-018-01023-3) and perform many rounds of both random and non-random cross-validations (https://doi.org/10.1007/978-0-387-39940-9_565, https://doi.org/10.1016/j.procs.2021.08.128) in order to prove their approach and the corresponding research results obtained are robust;

  • The Conclusions section is too big and not synthetic. A great part of it should be moved to Discussions.

Thank you for the opportunity to read and check this manuscript!>>

Dear Authors, Please carefully check and solve all English language and style observations as formulated (below) and sent to the editors of the Economies MDPI journal to which you submitted your manuscript:

<<Dear Editors,

  • In terms of English language and style issues - Grammarly (https://app.grammarly.com) on default settings (American English, Set Goals: Audience=Knowledgeable, Formality=Neutral, Domain=General) detected only for the text block resulting from the concatenation of Title+Abstract+Keywords+Conclusions: policy implications, limitations, further research: 25 correctness issues / critical alerts, and also 49 more complex ones / advanced suggestions.

Consequently, the resulting Grammarly overall/total score as reported by this online tool was just 79 (right edge of Fair i.e. >=70, but still not good i.e >=80, or Very Good/Excellent i.e. >=90) out of 100 (max) for this four-component sample above. Moreover, since the authors do not appear to be native English speakers, I suggest a comprehensive revision of the English language and style for the entire article using Grammarly or another specialized tool.

Thank you for the opportunity to read and check this manuscript!>>

Author Response

Author's’ Response to Reviewer 2

Dear Reviewer 2,

many thanks for giving us the opportunity to resubmit a revised version of our paper entitled "Youth entrepreneurship in Germany: Empirical evidence on the how, the why, the how many, the who and the when submitted to “Economies”.

Your comments helped us to improve the previous version. In the remainder of this letter, we respond to your comments (in italics) that were put forward along with our responses.

Best regards,

The Authors

Comments to the Author

The paper must follow all the instructions of the journal precisely indicated at: https://www.mdpi.com/journal/economies/instructions;

Response:

Again, we very carefully checked the complete manuscript and we are now very sure that we follow all instruction of “Economies”.

Comments to the Author

The chart in Figure 3 used the type dedicated to representing time series (e.g., Figure 2) although the variable on the X-axis is not represented by time or time stamps; In this particular case (Figure 3) the authors are trying to emphasize the relation between two variables indicated on those two axes. Therefore they should choose a scatter plot or similar, but not something dedicated to representing the evolution in time. Age is time-related but is not time;

Response:

Please excuse the misunderstanding. It is NOT our aim to document a time series within Figure 3. Rather, we want to show the entrepreneurial activity for different age groups. Instead of a line chart, we could also have used a column chart or a scatter plot, the message would remain the same. However, the line chart is much clearer, which is why we decided to use it and why we would like to keep it. We hope that this is comprehensible for you.

Comments to the Author

All references to equations/formulas must be explicitly and precisely formulated in the main text For instance, “... (see eq. N).” but not “as follows” or similar;

Response:

Thank you for the guidance and feedback, we have implemented the requested adjustments.

Comments to the Author

The authors must avoid ending some sections/subsections with formulas, figures, tables, or other components (e.g., Fig. 1 just before section 3, Table 6 just before section 4, and no explanatory text after). The authors are required to check the entire manuscript for similar issues;

Response:

We have done it the way we have done it in all other scientific journals we have published in so far: first, a figure or table is mentioned and explained in the text, then the table or figure itself appears. Of course, it may be that after the table or figure no further text follows/needs to follow in the corresponding chapter. In these cases, the table or figure is the end of the corresponding chapter and no further text follows before the heading of the next chapter. This does not affect the understanding of the text in any way. Nevertheless, in some places in the paper we have added some text after an illustration or table, if otherwise the next chapter heading would follow immediately.  

Comments to the Author

The authors must additionally ensure that all figures have the required resolution (min. 300 dpi, according to the Journal’s instructions at the link above). All three figures suffer in these terms;

Response:

Done. Figures 2 and 3 already fulfilled this criterion. However, we have adapted figure 1 accordingly, so it that has a 300 dpi resolution now, too. Many thanks for letting us know.

Comments to the Author

There is a high number of tables. I counted 9 in this manuscript. Some of them which are considered by the authors not essential for understanding the main flow of ideas in the manuscript must be moved to the Appendix section. If this section is not existing, the authors must create one;

Response:

Much appreciated. We have now created an appendix section and transferred four tables into this section, together with two newly created tables. Thus, the number of tables in the normal text has been reduced to 5, followed by six tables in the Appendix.

Comments to the Author

The List of Abbreviations at the end of the manuscript is needed (I counted at least 6);

Response:

Done.

Comments to the Author

The Limitations should be a dedicated section;

Response:

We do accept your suggestion in general as it is very often a reasonable one. However,

as we mention a very few limitations only a separate new section would be extremely short. Thus we would like to keep them included in the – now newly entitled – last section.

Comments to the Author

There are no details about the AUC-ROC accuracy values of the logit models obtained. The authors should present them precisely for each logit regression model;

Response:

This is a valuable comment, many thanks. We have now added two paragraphs on how we deal with multicollinearity. Furthermore, we have added one appendix table about the area under the ROC curve and about K-Fold Cross Validation Accuracy.

Comments to the Author

Each pair of predictors should be assessed in terms of collinearity: VIF against 1/(1-R^2) as indicated by Vatcheva et al.(2016) https://doi.org/10.4172/2161-1165.1000227&Freund et al.(2006) https://tinyurl.com/jypdexuc.

Response:

Many thanks, you are completely right. We have now added two paragraphs on how we deal with multicollinearity. Also, we have added one appendix table on one-dimensional generalized variance inflation factors and R² threshold.

Comments to the Author

The authors are also required to include more explanations and precise details about the standard view of the accuracy values (>=70% and <80%-fair models; >=80% and <90%-good models; >=90%-very good/excellent models) and their application here. They should provide more references to scientific papers where this topic is considered and the accuracy intervals are precisely defined;

Response:

Many thanks for this suggestion. We have added six additional references on the named issues in our reference list, related to the additional text.

Comments to the Author

At first glance, the final list of references seems reasonable. Still, I counted one conference paper that is not considered finalized research work and should be replaced with a reference to at least one journal paper;

Response:

We have done our best to find the results of the Conference paper in a more recent journal publication – and we succeeded.

Comments to the Author

Moreover, why discuss just the core results (section 5. Discussion of core results) and not all the results? The authors should precisely explain in the manuscript or rephrase;

Response:

Many thanks, we have rephrased it

Comments to the Author

The authors must understand that replicability as a fundamental principle in science (https://doi.org/10.1007/s10516-021-09610-2  https://doi.org/10.1038/nature.2016.20504 ) starts with data and it is not a fad but a necessity. Therefore, they should insert in the Data Availability Statement section at the end of the manuscript all precise links to all data providers’ / own datasets used when testing the entire approach;

Response:

Done. The original data stems from the publicly available database of the Global Entrepreneurship Monitor (GEM) website: gemconsortium.org

Comments to the Author

Following this scientific principle above, the authors should provide full details about the software and hardware they used for collecting, transforming, analyzing the data, and reporting the results presented in this manuscript (all data transformation/treatment and regression script sequences required if used);

Response:

While in our research fields, economic geography and entrepreneurship research, it is uncommon to inform journals or conference participants about the software and hardware used to produce empirical results, we are happy to let you know that we used the freely available version of R to analyse or – also freely available – GEM data.

Comments to the Author

The authors must also triangulate using many methods and techniques (https://doi.org/10.1038/d41586-018-01023-3) and perform many rounds of both random and non-random cross-validations (https://doi.org/10.1007/978-0-387-39940-9_565, https://doi.org/10.1016/j.procs.2021.08.128) in order to prove their approach and the corresponding research results obtained are robust;

Response:

We have done our best to apply some of these techniques and have added several remarks on this in section 3.2. For instance, K-fold cross-validation with 10 folds and 10 repetitions was performed, using the “Synthetic Minority Oversampling Technique” (SMOTE) as well as “Random Over-Sampling Examples” (ROSE). Additionally, Table F provides the mean area under the ROC curve (AUC) after randomly splitting the dataset into a training and testing dataset (80% training to 20% testing ratio) 100 times and applying the models. However, some of the weaknesses of the GEM data prevented us from applying all the techniques you have suggested.

Comments to the Author

The Conclusions section is too big and not synthetic. A great part of it should be moved to Discussions.

Response:

Many thanks, you are right: the final sections is not really a conclusion section. Instead, it shows implications, based on the main results and clearly structured into two parts: policy implications and research implications, and the research implications are mainly driven by the explained limitations of the paper. Consequently, we have renamed this chapter 6 and eliminated the term “conclusions”.  

Reviewer 3 Report

A very good paper. Some minor improvements to make are the literature cited to become more fit for argumentation of Hypotheses 1-3 (seems that the hypotheses by themselves eo exist 'in parallel' with the sources mentioned).

Fritisch and Wyrwych are arguing that the difference in entrepreneurship activity between regions in West and East Germany is related not to the Socialist past, but has deeper roots - contrary to your statement in the literature analysis.

Some a few places where either the proverb is missing or other minor omissions are made

Author Response

Author's’ Response to Reviewer 3

Dear Reviewer 3,

many thanks for giving us the opportunity to resubmit a revised version of our paper entitled "Youth entrepreneurship in Germany: Empirical evidence on the how, the why, the how many, the who and the when submitted to “Economies”.

Your comments helped us to improve the previous version. In the remainder of this letter, we respond to your comments (in italics) that were put forward along with our responses.

Best regards,

The Authors

Comments to the Author

“A very good paper. Some minor improvements to make are the literature cited to become more fit for argumentation of Hypotheses 1-3 (seems that the hypotheses by themselves eo exist 'in parallel' with the sources mentioned).

Response:

Many thanks for your encouraging overall statement. We have tried to connect the hypotheses more directly to the cited literature.

Comments to the Author

Fritsch and Wyrwych are arguing that the difference in entrepreneurship activity between regions in West and East Germany is related not to the Socialist past, but has deeper roots - contrary to your statement in the literature analysis.”

Response:

You are right, we have modified the respective part of the text.

Comments to the Author

Some a few places where either the proverb is missing or other minor omissions are made

Response:

Done, we now did our best to check for these aspects. The very careful and professional proof reading we have ordered also helped much.

Round 2

Reviewer 2 Report

Dear Authors,

You performed some changes.

I think your paper is now closer to being published.

I wish you all the best!

Minor changes required.